# The Current Status and Challenges of China Railway Express (CRE) as a Key Sustainability Policy Component of the Belt and Road Initiative

Kyoung-Suk Choi

Department of International Trade, Jeonbuk National University, Jeonju 54896, Korea; koyaku@jbnu.ac.kr

**Abstract:** Under the auspices of the Belt and Road Initiative (BRI), China has been expanding the operation of its CRE (China Railway Express) system that links China and Europe. The CRE is today regarded as an important measure of progress by which BRI-related goals are achieved, and it has the potential to become the most sustainable mode of long-range transport. The system has been growing rapidly thanks to the active support of the Chinese government. As the Eurasian trade and logistics environment changes, CRE will become increasingly important as a third option that complements air and sea transport, with the demand continuing to accelerate among shippers for CRE service. Despite the expanding role and potential importance of the CRE system, few academic studies have been undertaken on the relevant CRE policies and status, especially in international academic journals. Thus, this study comprehensively reviewed the specific operation status of CRE system by route and region, and identified specific challenges that need to be addressed in order to continue its sustainable development.

**Keywords:** China Railway Express (CRE); Belt and Road Initiative; block train transportation service; railroad policy; railway networks





## 1. Introduction

Block train transportation service (BTTS) is a chartered train transport service consisting of about 50 trains per unit that directly links a rail transport system to the shippers, allowing for non-stop delivery to the destination. BTTS is mainly used to transport containers [1] and is operated according to fixed schedules [2]. In China, BTTS is referred to as China Railway Express (CRE) in English, and these systems are in wide use to connect China with Central Asian and European countries. Since CRE launched in 2011, China's BTTS has experienced numerous noteworthy developments. While CRE grew relatively slowly from 2011 to 2013, from 2014 onwards, the Belt and Road Initiative (BRI), China's national vision program, has resulted in a sharp increase in the number of operations and the volume of goods transported.

First proposed by President Xi Jinping in 2013, the BRI aims to build a far-reaching Eurasian economic community that encompasses all nations situated along the two Silk Roads (the Silk Road Economic Belt and 21st-Century Maritime Silk Road). The five-stage connectivity plan guides policy related to infrastructure, international trade, finance, and people. The BRI treats railways as key infrastructure for accomplishing the achievement of a Silk Road Economic Belt, and block train operation using the Eurasian intercontinental rail network has been identified as a core task to be promoted; progress in this area is even treated as a performance metric [3,4]. Pursuant to the BRI, the Chinese government is currently promoting the cooperating international linking of railway infrastructure and encouraging local governments to actively expand CRE operations. As a result, each local government now competes for the CRE operations to secure a strategic position as a railway logistics hub.

With such enthusiasm by central and local governments, CRE uses breaks records every year. As of 5 November 2020, 10,180 trains were operating between China and Europe, for a total transport cargo volume of 9277 thousand TEU, up 54% from the previous year [5]. Most CREs are managed by China Railway, a national enterprise that operates trains between, among others, such as Chongqing–Europe (Yuxinou), Zhengzhou–Europe (Zhengou), Wuhan–Europe (Hanxinou), and Chengdu–Europe (Rongou). As of 2020, CRE connects 92 cities in 21 countries [5,6].

Maersk, the world's top container liner company, recently began providing a new multimodal transportation service, using block trains connecting East Asia and Europe [7]. Pantos, a representative the logistics company in Korea, also has recently expanded BTTS, using TCR and TSR on its Korea–Europe transport routes. It is predicted that BTTS use will continue to grow as regular services operating these block trains become more predictable and entrenched [8]. From the viewpoint of logistics companies and shippers, BTTS is rapidly becoming a favored alternative transport system, appreciated for its special characteristic as a single mode of a multimodal transport system, while the Chinese government continues to believe that BTTS has the potential to become the most efficient and sustainable long-haul transport system between China and Eurasia.

Despite the growing popularity of this system, few researchers have focused on CRE-related systems, with only a few research studies having been conducted almost exclusively in China. This study was therefore undertaken with the following three research objectives: (1) investigate the current operating status of the CRE, along with the Chinese government's policy related to CRE development under the BRI; (2) identify problems arising as a result of the rapid increase in CRE operation; and (3) propose the challenges that must be addressed for the sustainable development of CRE, especially as a major transport system, especially in trade with East Asia, Middle Asia, and Europe.

This study was based on a comprehensive review of all publicly available CRE-related materials and data, including articles in Chinese and English journals, as well as government documents. The information thus revealed may serve as a reference for more efficient and rational Eurasian transport networks. It is also expected that this study will stimulate additional research, particularly English-language research, into CRE and block trains.

## 2. Literature Review

To date, most CRE-related research has been performed and published exclusively in China, with only a few relevant studies appearing in international journals. The Chinese research has mostly taken the form of comprehensive literature reviews, and not empirical studies, with the explicit goal of expanding CRE operation in central and local governments [3,9–14].

In China, Geng's [15] research focused on the linkages formed between China and neighboring countries under the BRI. Geng ultimately proposed flexible government subsidies to support the CRE, the establishment of a CRE main logistics hub, and the acceleration of improvements to service quality through the establishment of a logistics information system. Ma [16] began with the idea that the CRE is a means of building cooperation with neighboring countries, accelerating economic reform and openness in China and promoting the BRI, and, from there, investigated the status and challenges associated with the CRE, as well as optimal paths to its completion.

Additional Chinese research, including that by Chen et al. [12], has empirically analyzed CRE's impact on China-EU trade by performing a double differential regression analysis on the calculated gravitational coefficient of China and other countries around the world, from 2015 to 2019. Zhao and Guo [9] also applied the gravity model and verified that the CRE's western corridors were likely to promote international trade with BRI countries. Fu et al. [13] built a multi-purpose optimization model to calculate the realistic cost and time of a CRE departing from Wuhan, Zhengzhou, Chongqing, and Changsha (all in western regions of China) and analyzed the CRE's economic feasibility and low

loading rate problem, eventually proposing several optimized alternatives designed to preempt issues.

Outside of China, only a few studies have reviewed the CRE as part of other continental railway and multimodal transport systems [17,18]. Choi [17] emphasized the policy establishing a unified rail transport system that connects Korea–China train ferries and the CRE to handle South Korean trade with Eurasian countries that passes through China. Choi highlighted the need for Korean companies to improve communication and cooperation with local Chinese governments, CRE platform companies, and rail-related organizations and called for investment and other government support to incentivize logistics companies to participate in the BTTS market.

The CRE-related studies that have appeared in international journals broadly fall into two categories. In the first category are those articles that examine the economics, speed, efficiency, and competitiveness of BTTS and CRE from the perspective of Asia–Europe connections [19–21]. Rodemann and Templar [20] highlighted Eurasian rail transport as potentially competitive with existing sea and air transport routes. These authors examined factors that have hindered the development of a Eurasian rail transport system to date and suggested measures to overcome them. Besharati et al. [21] analyzed the current operating situation for the sustainable development of the China–Europe block train and calculated the economical transport cost, suggesting means of the efficient use of empty containers coming from Europe due to imbalances in import and export cargo between China and Europe countries. Li et al. [22] designed and analyzed a multi-logit model based on a generalized cost function to evaluate and compare the CRE's market competitiveness with sea transport between China and Europe.

A second group of researchers have focused on CRE logistics hubs and facilities, including connections, routes, related hinterland, and logistics centers, etc., and employed empirical tests [19,23,24]. Among them, Zhao et al. [23] examined CRE operations and evaluated the importance of cities as a consolidation center through a Technique for Order of Preference by Similarity to Ideal Solution (TOPSIS) model. Ultimately, five cities (Taiyuan, Xi'an, Zhengzhou, Wuhan and Suzhou) were identified as ideal consolidation centers. Jiang et al. [19] analyzed the pattern of the CRE hinterland, and compared and analyzed the five routes from the viewpoint of sea transport and freight rates. Yang et al. [25] estimated the CRE's impact on the Chongqing–Europe transport route, especially the impact of the outbound transport network, and compared it to the impact of maritime transport. This paper also evaluated CRE's impact on the formation of the Chongqing International Logistics Center. Yin et al. [26] analyzed the inland railway container center station for CRE.

A few researchers have attempted to investigate research trends and identify future research agendas related to the BRI by comprehensively reviewing journal articles and undertaking keyword frequency analysis. These studies determined that the BRI has a positive impact on the transport industry, specifically improved logistics capabilities, strengthened connectivity between transport networks, and superior intermodal transport. As a result of keyword analysis, "transportation" and "infrastructure" have been identified as the core keywords in relation to BRI [27,28]. Hyun and Kim [29] examined the betweenness centrality of words appearing in BRI-related Korean journal articles and identified "railroad" as the most important keyword.

Based on our comprehensive review of the CRE-related literature, including Chinese-language literature, it was confirmed that, to date, no comprehensive overview of the CRE has yet been offered in English. This study is an attempt to fill this research gap.

As a key component of the BRI, the CRE has experienced rapid growth. This has led to a number of problems. This study is therefore significant as a multidimensional follow-up that provides an interim check on the operation of the CRE and on the research agendas derived from various problems.

## 3. CRE-Related Policies and Operation Status

### 3.1. The Relationship between the BRI and China Railway

President Xi Jinping first mentioned the Belt and Road Initiative (BRI) during his visit to Kazakhstan and Indonesia in September and October 2013. This was followed in 28 March 2015 by the release of the official outline for the BRI, jointly issued by the National Development and Reform Commission (NDRC), the Ministry of Foreign Affairs (MOFA), and the Ministry of Commerce (MOFCOM) of the China. Today, the BRI represents China's top priority. The BRI is long-term policy scheduled to be completed by 2049, a year which coincides with the 100th anniversary of the formation of the Chinese Communist Party. The BRI aims to build and strengthen cooperation, connectivity, and economic win-win partnerships with countries along the two identified Silk Roads (land and sea). The BRI encompasses 92 countries and regions, represents more than a third of the world's GDP, and encompasses within its ambitions 2/3 of the global population [18,30,31]. As shown Figure 1, the ultimate goal of the BRI is the economic integration of Asia, Eastern Africa, Eastern Europe, and the Middle East, a region mainly composed of emerging markets.

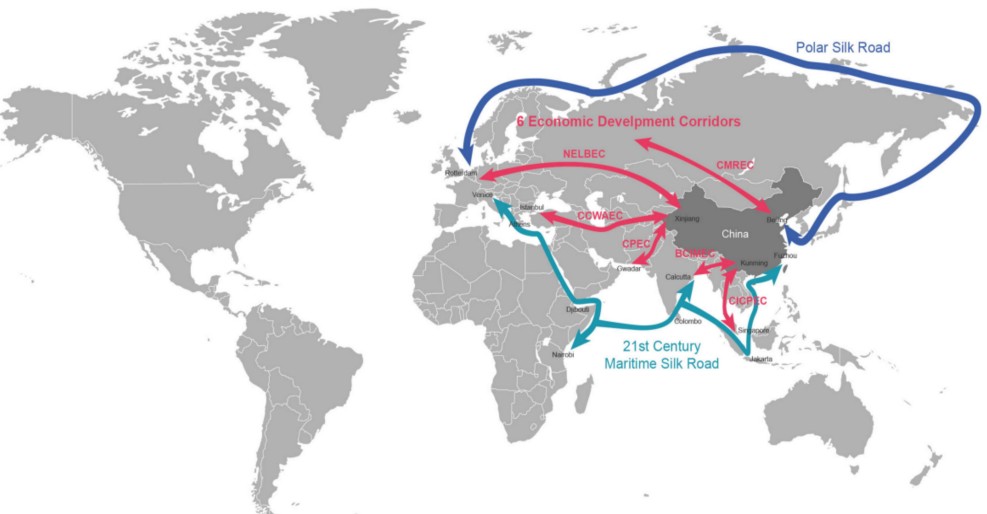

**Figure 1.** The geographic area encompassed by the BRI. Source: Belt and Road Initiative [32].

This goal is to be achieved by five strategies, the development of supportive policies, infrastructure, international trade finance connections, and public sentiment [31]. The five strategies was summarized in Figure 2. Among these, the "infrastructure" strategy, which calls for the construction of new roads, railroads, airports, and ports among the BRI countries, accounts for approximately 68% of the total investment to be made pursuant to the BRI [17,18,33]. In other words, the BRI represents one of the most ambitious infrastructure investment plans in the history of the words, and it has tremendous potential to stimulate regional economic growth in Asia, Europe, and Africa [24].

Since the announcement of the BRI, China has attempted to improve and expand its railroad network within the framework established by existing trans-Eurasian railroad routes. A China–Laos Railway, China–Thailand Railway, Jakarta High-Speed Railway, and Hungary–Serbia Railway, all of which are part of a Pan-Asian Railway Central Line, are all currently being built. As part of this infrastructure push, China is currently expanding block train services for the transportation of cargo between the BRI countries. The CRE is gradually becoming a key performance indicator by which success at achieving the ambitions of the BRI is measured.

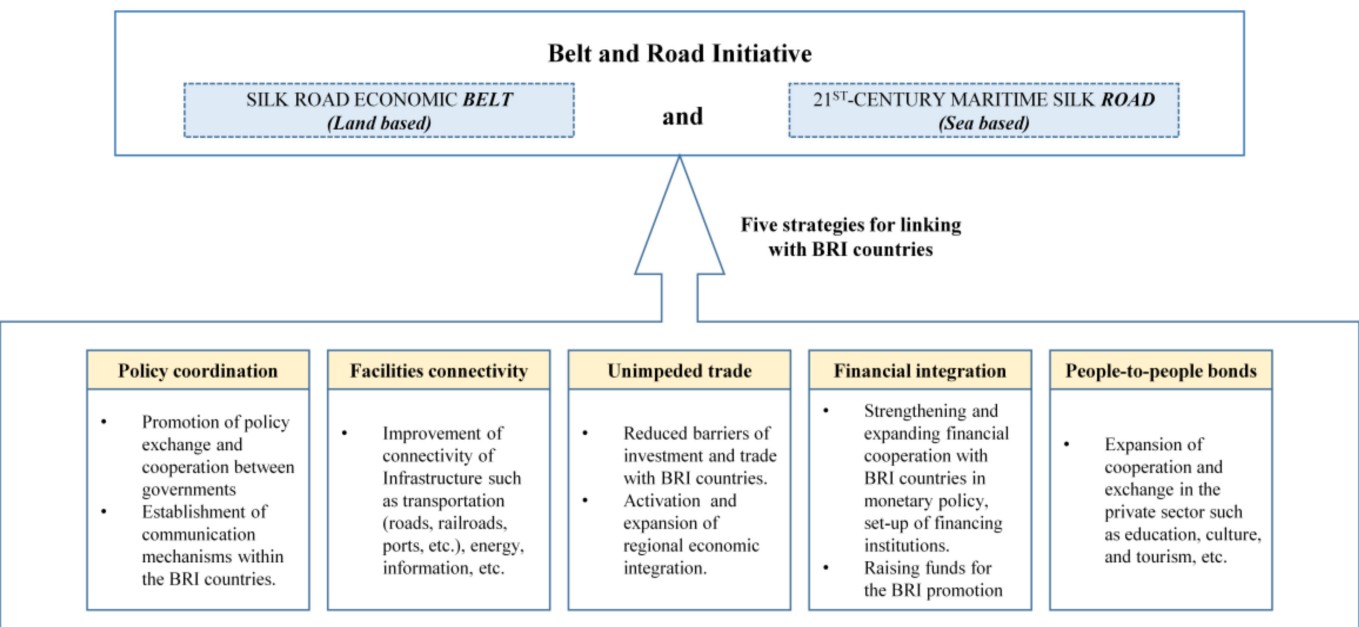

**Figure 2.** The five strategies by which linkage among BRI countries will be achieved.

### 3.2. CRE Policies under the BRI

China has been working to improve the operational efficiency of the nation's railway network as part of the 12.5 (12th 5-year plan: 2011–2015) and 13.5 (13th 5-year plan: 2016–2020) national economic development plans. Outside of the BRI framework, China has made efforts to develop the Silk Road economic belt by improving railroad networks within Eurasia. To broaden the role and function of railways, the Chinese government has significantly expanded the linkage and operating distance of the nation's railroads, and the China Railway Corporation has been extensively reformed, experiencing a reorganization of its freight transport system, standardization of its railroads, and continuous improvements to its operational efficiency through technological innovation.

The most critical government related CRE program is referred to as the "Vision and Actions on Jointly Building Silk Road Economic Belt and 21st-Century Maritime Silk Road" program, which was jointly announced in March 2015 by the NDRC, MOFA, and MOFCOM. That document highlighted the importance of CRE for land rail networks [23]. A second policy, the Mid-to-Long-Term Railway Network Plan (MLRNP), was ratified and published by the NDRC within the Ministry of Transport, which led to the 2016 establishment of the China Railway company to promote the sound and orderly development of CRE between China and Europe.

The MLRNP sets out four main goals. The first is to serve as the basic rail network framework. This aspect of the document includes a vision of the further acceleration of the construction of domestic Midwestern region railways and international railway networks at the basic stage, and the expansion of high-speed rail networks. The second is improved rail transport service by developing the capacity of cargo transportation services, organically establishing connectivity with logistics hub regions, and continuous improvements to the level of technology employed and information gathered. The third is to ensure independent technology development capabilities and core competitiveness through innovation and system integration, enabling smooth CRE operation in high-speed, highland, and alpine regions. The fourth is the separation of government and private enterprises and achieving the reform of the investment and funding system, including the land development, establishment and approval of railroad development foundation, and social capital investment frameworks. It also mandates that central and local governments strengthen support for railroad construction [34].

On 19 October 2016, the "Leading Group for the Construction of BRI" was formed to promote systematic construction of the CRE. Related divisions under the leadership of the Central Communist Party, the State Council, the National Development and Reform Commission, and the China Railway Corporation jointly announced the CRE construction development plan (2016–2020) [35].

In April 2017, the relevant railway regulatory authorities of seven countries, including China, Belarus, Germany, Kazakhstan, Mongolia, Poland, and Russia, signed the "Regarding deepening the CRE Cooperation Agreement" between China and Europe, a landmark event acknowledged as a major achievement of the BRI [36]. Subsequently, in May 2017, China formed the CRE Transport Coordination Committee to provide a body through which China and European parties could work together to strengthen cooperation, attract broader participation, and ensure the efficient and stable operation of the CRE [37]. In December 2017, the NDRC issued the "Notice on Deepening the Market Reform of Railway Freight Transport Prices and Other Related Issues", the main purpose of which is to expand the range of rail freight weights that rail companies are permitted to charge for each product from 10% to 15% [38]. Table 1 shows a more specific CRE drive process.

This series of railroad-related policies can be seen as a strategic attempt to actively respond to the attraction of rail cargoes and to establish a consistent rail transport system between BRI countries. Through policy efforts like these, the CRE will continue to develop.

### 3.3. Operation Status of CRE Borders

According to the China Railway Container Transport Corporation (CRCT), the cargo transport performance along CRE railways has shown remarkable growth, more than doubling its transportation capacity since 2014. Of particular note is the growth in cargo imports from Europe, which were essentially nonexistent in 2014 but show up to 323,000 TEU in 2019, at which point they made up 44.5% of the total shipping. Today, cargo containing home appliances, electronics, clothing, auto parts, and machinery are mainly transported using CRE, and with CRE emerging as the third most-popular method of transport between China and Europe after sea and air [12,37]. The CRE has three routes between China and Europe: Eastern, Central, and Western. The eastern routes mainly attract import and export goods from the east, south, and north coastal areas of China and Europe, while connecting Russia, Belarus, Poland, and Europe via the Manzhouli railway border. The Central routes are focused on cargo transported between North/Central China and Europe. It enters Europe at the Erenhot railway border and travels through Mongolia, Russia, Belarus, and Poland. The Western routes mainly carry goods between the central and western regions of China and Europe, and is connected to Europe at the Alashankou (Khorgos) railway station, from whence it proceeds to Kazakhstan, Russia, Belarus, and Poland [2]. Figure 3 shows CRE routes through major railway borders and CRE-related cities in more detail.

As can be seen in Table 2, these three routes pass through either Manzhouli, Erenhot, or Alashankou (Khorgos), depending on the origin of each train and its cargo. Alashankou is the most active gateway with most of the block trains passing through, followed by Manzouri, Erenhot, and Khorgos, in that order [6,16] (Table 3).

**Table 1.** The CRE drive process.

| Phase | Year | Date | Contents |
|---|---|---|---|
| Beginning (active exploration) | 2011 | 19 March | The CRE commences operation from Chongqing in China to Duisburg in Germany (16 days). |
| | 2012 | 1 August | The "Yuxinou (Chongqing–Europe route)" seminar focuses on facilitating customs clearance, with the participation of eight countries. China, Russia and Germany reach consensus on further simplifying the customs clearance process and implementing the principle of mutual supervision. |
| | | 24 October | The Wuhan–Czech corridor pilot operation. |
| Building and expansion | 2013 | 18 March | The first return test train of "Yuxinou" from Duisburg to Chongqing. |
| | | 29 March | Chinese President Xi Jinping visits the port of Duisburg to observe the arrival of the Chongqing-Duisburg CREs. |
| | 2014 | 14 August | The first CRE coordination meeting held in Chongqing, discussing how to unify brand logo, transport organization, wholesale pricing, service standard, management team, and coordination platform. Promulgated the "Interim Measures for CRE Organization Management", and signed the "Establishing CRE Domestic Transportation Coordination Meeting Memo". |
| | | 18 November | The first "Yixinou (Yiwu–Europe route)" departs from Yiwu to Madrid, Spain. |
| | 2015 | 28 March | The Chinese government issues "Vision and Actions for Promoting the Joint Construction of the Silk Road Economic Belt and the 21st Century Maritime Silk Road" which calls for the establishment of a CRE railway transport corridors, port customs clearance coordination mechanism, and transportation channels that connect domestic and foreign countries. |
| Active development | 2016 | 14 April | New route departs from Dongguan, passing through Manzhouli, Russia, Belarus, and Poland, and arriving in Duisburg, Germany (13,000 km, the longest CRE in China). CRE operating companies in Xinjiang, Chongqing, Zhengzhou and other cities established the China–Europe international freight transportation CRE alliance in Urumuchi. |
| | | 15 April | The "Xinjiang Declaration" unifies CRE operating mechanism and optimizes the transportation organization and spatial layout of the CRE. Unified CRE brand officially launched. |
| | | 8 June | President Xi Jinping and Polish President Duda attend the ceremony for the CRE's first arrival in Poland. |
| | 2017 | 8 October | Release of "CRE Construction and Development Plan (2016–2020)" under the BRI. |
| | | 20 April | Seven countries, including China, Belarus, Germany, Kazakhstan, Mongolia, Poland and Russia, sign the "Regarding deepening the CRE Cooperation Agreement". |
| | | 1 May | Transport document form unified with Germany, France, and other countries |
| | | 26 May | The CRE transport coordination committee established with seven CRE platform companies in Chongqing, Chengdu, Zhengzhou, Wuhan, and other regions. The international multimodal transport information platform and the CRE refrigerated container information-sharing platform are established. |
| | | 18 November | The total number of CRE operation exceeds 6000. |
| | | 20 December | CRE trial from Chengdu to Tilburg in the southern Netherlands via Urumuchi. |
| | 2018 | 15–16 October | The NDRC holds the CRE meeting in Chongqing to make improvements in the CRE mechanism and enact new rules and laws. |

**Table 1.** *Cont.*

| Phase | Year | Date | Contents |
|---|---|---|---|
| Stable improvement | 2019 | 22 April | "The Belt and Road Initiative: Progress, Contribution and Prospects" is announced, emphasizing the role of the CRE in promoting multi-country cooperation as an international train operation mechanism. By the end of 2018, CRE connects 108 cities in 16 countries on the Eurasian continent. The number of the CRE operation exceeds 13,000, transporting more than 1.1 million TEUs. Customs clearance time reduced by 50% as a result of customs clearance agreements with the BRI countries. |
| | 2020 | 8 March | The NDRC announces round trip rate of over 90%. |
| | | 3 April | The MOFCOM issues the "Notice on Further Utilizing the Role of the CRE to respond to the COVID-19 and Stabilizing Foreign Investment Promotion Fees", proposing 11 specific measures and work requirements. |
| | | 14 April | CRE (Wuhan) X8015 arrives at the Duisburg in Germany under above measures. |
| | | 5 June | CHINA RAILWAY announces that in May this year, the number of the CRE operation have exceeded 1033, increasing 43% over the previous year. The transported cargoes reaches 93,000 TEUs, increasing 48% over the previous year. |
| | | 8 November | The X8020, jointly organized by CRE platform companies in Yiwu, Chongqing, Zhengzhou, Xi'an, and other cities, begins first cross-border e-commerce CRE from Yiwu to Brussels. |

Sources: author summary based on Belt and Road Portal [32].

**Table 2.** China's major rail transit routes.

| Routes | Corridors | Main Railway Border | Main Supply Regions |
|--------|-----------|---------------------|---------------------|
| Eastern | Beijing–Harbin–Manzhouli–Russia–Belarus–Poland–other European countries | Manzhouli | East China, South China and North China |
| Central | Erenhot–Mongolia–Russia–Belarus–Poland–other European countries | Erenhot | Central China and North China |
| Western | Alashankou (Khorgos)–Kazakhstan–Russia–Belarus–Poland–other European countries | Alashankou Khorgos | Southwest, Northwest |

Source: Author summary.

**Table 3.** Number of CRE operations at major railway borders. Unit: train.

| Railway Border | 2018 (1–11) | Cumulative Total |
|----------------|-------------|------------------|
| Erenhot | 1000 | 1750 |
| Manzhouli | 1074 | 3000 |
| Alashankou | 2388 | 7123 |

Source: Sirponeer [38].

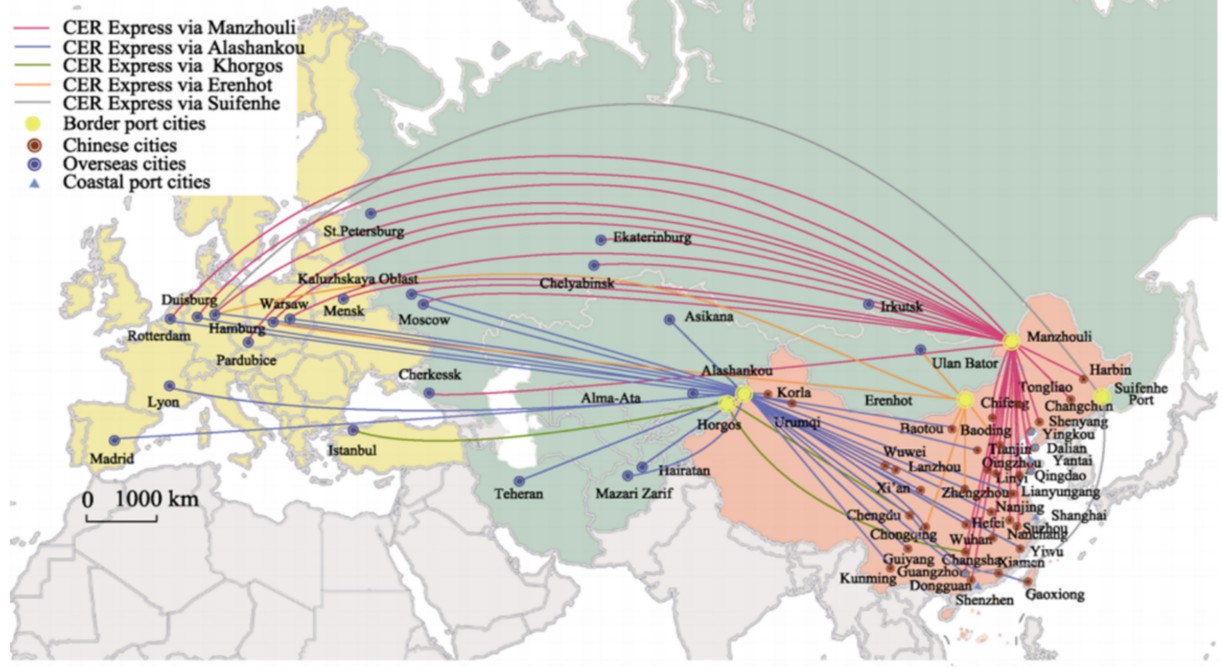

**Figure 3.** CRE routes and cities based on OD (Origin and Destination). Source: Wang et al. [39], p. 1279.

Alashankou, where 70% of the CRE is concentrated, used to be severely congested and bottlenecked by customs clearance and transshipment problems. However, after the completion and opening of the Khorgos border gate in 2016, cargo concentrated in Alashankou began to be dispersed to Khorgos, gradually alleviating the bottleneck. Despite this improvement, the bottleneck nevertheless remains serious.

### 3.4. The CRE Operation Status by Routes

The CRE is an important instrument for economic cooperation with BRI countries, as being a direct means of effectuating the construction of the BRI. Since its first opening in 2011, the CRE has grown at a rapid pace, with new BTTSs launched in many inland and coastal port cities [32,38]. Table 4 shows the number of round-trip operations and amount

of container cargo volume of the CRE between 2011 and 2019. As seen in Table 4, from 2011 to 2013 inbound cargo was zero, while today inbound CRE cargo is known to have steadily increased since 2014. Nevertheless, the imbalance between inbound and outbound cargo transported by CRE remains high.

**Table 4.** CRE operation status (2011–2019). Unit: ten thousand TEU.

| Year | Classification | Export and Import | Total | Year | Classification | Export and Import | Total |
|------|----------------|-------------------|-------|------|----------------|-------------------|-------|
| 2011 | # of operations | 17 outbound<br>0 inbound | 17 | 2016 | # of operations | 1130 out<br>572 in | 1702 |
|      | # of Containers | 0.14 out<br>0 in | 0.14 |      | # of Containers | 9.7 out<br>4.3 in | 14 |
| 2012 | # of operations | 42 out<br>0 in | 42 | 2017 | # of operations | 2399 out<br>1274 in | 3673 |
|      | # of Containers | 0.37 out<br>0 in | 0.37 |      | # of Containers | 21.2 out<br>10.6 in | 31.8 |
| 2013 | # of operations | 80 out<br>0 in | 80 | 2018 | # of operations | 3696 out<br>2667 in | 6363 |
|      | # of Containers | 0.70 out<br>0 in | 0.70 |      | # of Containers | 31.9 out<br>22.3 in | 54.2 |
| 2014 | # of operations | 280 out<br>28 in | 308 | 2019 | # of operations | 4525 out<br>3700 in | 8225 |
|      | # of Containers | 2.39 out<br>0.23 in | 2.62 |      | # of Containers | 40.2 out<br>32.3 in | 72.5 |
| 2015 | # of operations | 550 out<br>265 in | 815 | | | | |
|      | # of Containers | 4.7 out<br>2.2 in | 6.9 | | | | |

Source: CRCT [37].

As of January 2019, the CRE departed from 43 cities in China. In particular, Chengdu and Chongqing, representative cities of the Midwest of China, operated the CRE twice a day on average, and Zhengzhou, Wuhan, and Xian operated the CRE more than once a day. Dalian, Yingkou, Tianjin, and Lianyungang, coastal cities, also operate the CRE once a day on average. The CREs departing from these regions engage in the multimodal transportation of containers of cargos collected from Japan and South Korea, as well as local cargo collected from each region of China [40]. Figure 4 shows the CRE related cities as of 2018. CRE routes have today been actively expanded to include various cities, such as Chongqing, Hefei, Suzhou, Chengdu, Wuhan, Zhengzhou, and Changsha, etc., with different CRE names being used depending on the route and region.

Table 5 summarizes route, number, volume, primary cargo, distance, transit time, and departure time information associated with each of the six major CRE routes.

As of 2016, Rongou departing from Chengdu accounted for the highest market share of goods transported (16.5%), followed by the Yuxinou routes departing from Chongqing (16.3%). The largest number of CREs operate in three regions: Chengdu, Chongqing and Xi'an, accounting for about 65.68% of the total market [19]. Changanhao in Xi'an achieved the most rapid growth compared since 2017, with 68.7% of goods destined for Europe are goods such as textile materials, textile products, machinery and parts, and plastics. Conversely, cargo such as automobiles, auto parts, and wood from Europe to Xi'an are imported using mainly the CRE [43]. Table 6 below shows the growth trend of CRE since its first operation in 2011, focused on the six major cities.

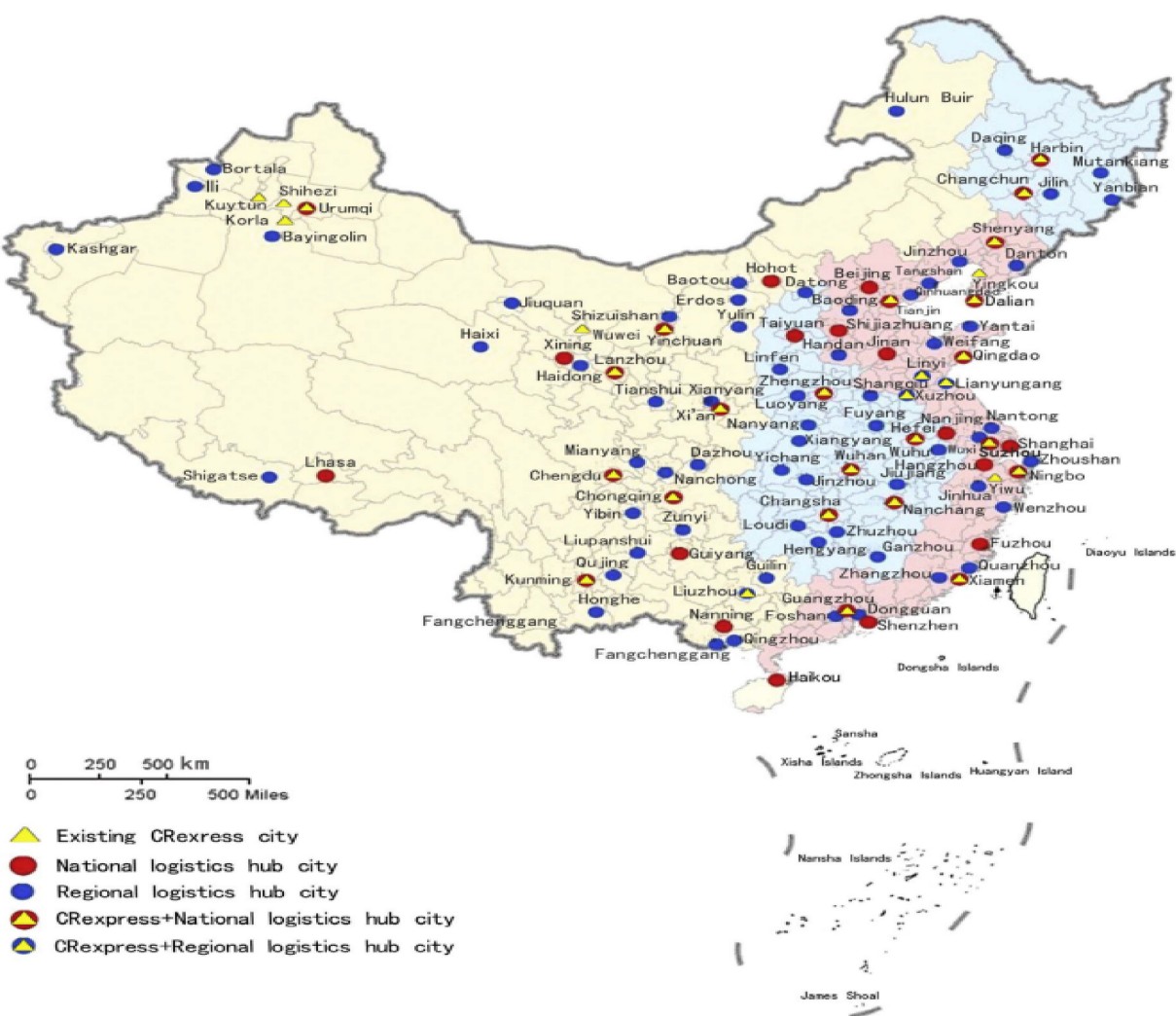

**Figure 4.** CRE-related countries or regions. Source: Zhao et al. [23]., p. 61.

**Table 5.** Summary of major CRE routes.

| CRE Name | Main Cargoes | Round Trip (Between China and Europe) | Corridors | Distance and Time | Departure Time | Cargo Collection |
|---|---|---|---|---|---|---|
| Xiangou (Changsha–Europe) | IT products, textiles, ceramics, tea, construction machinery, medical instruments | Outbound | Changsha–Erenhot–Warsaw–Hamburg | 11,808 km/13–15 days | Every Saturday | Cargo collection on average 3 days before round trip |
| | | Inbound | Hamburg–Warsaw–Alashankou–Hangsha | | Every Saturday | |
| Zhengou (Zhengzhou–Europe) | IT products, clothing, auto accessories, construction machinery, medical instruments | Outbound | Zhengzhou–Alashankou–Kazakhstan–Russia–Belarus–Poland (Marasevic Station)–Hamburg | 10,461 km/13 days | Every Monday–Saturday | Cargo collection on average 3 days before round trip |
| | | Inbound | ① Hamburg–Marasevic–Alashankou–Zhengzhou ② Hamburg–Marasevic–Erenhot–Zhengzhou | | Every Monday, Tuesday, Thursday Every Wednesday, Friday, Saturday | |
| Yixinou (Yiwu–Europe) | Bags, stationery, art and crafts, daily necessities | Outbound | Yiwu–Alashankou–Marasevic–Duisburg–Madrid | 13,052 km/21 days | Every Wednesday | Cargo collection on average 2 days before round trip Transshipment at the French and Spanish border |
| | | Inbound | Madrid–Duisburg–Marasevic–Alashankou–Yiwu | | Every Friday | |
| Hanou (Wuhan–Europe) | IT products, machinery, chemical products | Outbound | ① Wuhan–Arasankou–Marasevic–Duisburg | 10,880 km/15 days | Every Wednesday | Cargo collection on average 6 days before round trip |
| | | | ② Wuhan–Alashankou–Marasevic–Hamburg | | Every Friday | Cargo collection on average 4 days before round trip |
| | | Outbound | Duisburg–Marasevic–Alashankou–Wuhan | | Odd Week: Friday Even-numbered weeks: Friday, Saturday | Cargo collection on average 5 days before round trip |
| Rongou (Chengdu–Europe) | IT products, food, daily necessities, auto parts | Outbound | ① Chengdu–Alashankou–Lodz ② Chengdu–Alashankou–Lodz–Nürnberg ③ Chengdu–Alashankou–Lodz–Tillberg (Netherlands) | 9826 km/14 days | Every Thursday and Saturday Every Wednesday Every Wednesday | Cargo collection on average 3 days before round trip |
| | | Inbound | ① Tilberg–Lodz–Alashankou–Chengdu ② Nuremberg–Lodz–Alashankou–Chengdu | | Every Tuesday Every Friday | |
| Yuxinou (Chongqing–Europe) | IT products, auto parts | Outbound | Chongqing–Alashankou–Marasevic–Duisburg | 11,000 km/14–15 days | Every Monday, Thursday, Saturday Every Monday, Thursday, Saturday | Cargo collection on average 3 days before round trip |
| | | Inbound | ① Duisburg–Marasevic–Alashankou–Chongqing ② Duisburg–Marasevic–Erenhot–Chongqing | | Every Friday | |

Remarks: ① reservation required 10 days in advance; ② one-way operation to Germany takes about 15 days; ③ normally FEU containers are used; ④ passing Erenhot route takes approximately 2 more days than Alashankou; ⑤ transshipments are required in China, Kazakhstan, Russia, and Belarus. Source: References [19,41,42].

**Table 6.** Number of CRE operation by major city (2011–2018). Unit: train.

| Major City (Node) | 2011–2013 | 2014 | 2015 | 2016 | 2017 | 2018 |
|---|---|---|---|---|---|---|
| Chongqing | 96 | 102 | 257 | 420 | 700 | 1442 |
| Chengdu | 32 | 45 | 103 | 453 | 777 | 1587 |
| Zhengzhou | 13 | 77 | 156 | 251 | 493 | 752 |
| Wuhan | 1 | 26 | 164 | 122 | 375 | 417 |
| Hefei | 0 | 13 | 28 | 54 | 70 | 182 |
| Wrumuqi | 0 | 0 | 0 | 135 | 710 | 1000 |

Source: Zhao and Yang [44]. p. 67.

## 4. Challenges for Sustainable CRE Development

Over the past several years, the CRE train service between China and Europe has been growing rapidly in terms of quantity, but many problems remain, including border bottlenecks, inefficiencies associated with customs clearance, the imbalance of round-trip cargoes, and the intensification of competition. This section investigates these specific challenges and considers how stable and sustainable growth within the Eurasian continent can be achieved.

### 4.1. Resolving Cargo Concentration and Bottlenecks at CRE Borders

Of the four relevant borders, the overconcentration of CRE trains at Alashankou is particularly serious. Due to the difference in gauges used between TCR (Trans–Chinese Railway: standard gauge) and TSR (Trans–Siberian Railway: broad gauge), transshipment is essential at borders like this to ensure the smooth operation of the China–Europe CRE. This high concentration of CRE cargoes means that logistics challenges are common, such as increased demand for transshipment equipment and longer transit times. To address these problems, China should quickly develop alternatives that mitigate railroad congestion.

As mentioned above, there are four CRE borders along which customs procedures and transshipment are carried out. However, the waiting time for cargo differs from border to border, and it may take anywhere from one day to a week for cargo to move on. In the case of the Yiwu– or Chongqing–Madrid routes, CRE transit time should take no more than 16–18 days, but in reality it often takes 22 to 30 days as cargo waits for transshipment. Passing through Erenhot takes two days longer than Alashankou. Of course, these problems affect the entirety of the CRE operation schedule, as ultimately the network is unable to adequately satisfy the increasing demand [1,16,45]. The need to standardize and simplify transshipment and customs clearance procedures at these borders remains urgent.

The lack of trains or transshipment equipment at railway borders has exacerbated these delays and hindered the efficiency of the CRE. Ultimately, sufficient number of appropriate freight trains must be secured, and the overall level of logistics infrastructure at CRE borders must be improved, if the growth of CRE is to be sustainable and competitive as a form of international transportation.

At the same time, the borders passing through each route by regions must be properly rearranged. In particular, there is a need for an alternative to more actively disperse Alashankou's cargoes to Khorgos and Erenhot by developing and improving the logistics infrastructure at two borders. In addition, it should also improve information sharing, so that shippers can check border waiting times in real time.

As the CRE passes through many Eurasian countries customs procedures affect the competitiveness of international transportation. Unfortunately, diverse customs clearance standards are applied in different countries, which lowers the overall operational efficiency of the CRE. China accordingly recently established the CRE Transport Coordination Committee to create a forum for deliberation and hopefully coordination through which relevant parties from China and Europe could promote multilateral cooperation. The Committee, however, cannot change the underlying reality that countries have diverse customs practices due to their unique trade barriers and regional protectionism pressures,

and effective coordination has not yet been achieved. Coordination challenges also exist within China, diverse parties, including the central and local governments, enterprises, domestic carriers, and the CRE platform companies, may have different and conflicting recommendations [16,45].

It is recommended that the government quickly establish a common clearance platform to standardize and simplify customs clearance through stronger governance partnerships with the BRI countries. To promote the convenient, simplified, and quick customs clearance, the government may consider designating other inland borders where customs clearance would be possible in addition to the four borders already in use.

### 4.2. Alleviating Competition and Subsidies between Local Governments

CREs are recognized as important logistics infrastructure that expand international trade and investment along BRI's two Silk Roads [1]. In recent years, as local governments increasingly acknowledge the uptick in the number of CRE operations and cargo volume as a primary achievement of the BRI, they have competed secure positions as rail transport hubs. The competition is intensifying daily, particularly among cities in the mid-western regions, Chongqing, Chengdu, Wuhan, Xian, Zhengzhou, Hefei, and Lanzhou. To attract cargo, these local governments have worked to reduce the transport freight gap with sea transportation, and have been providing financial support to shippers that use the CRE. The subsidy amounts vary slightly by province and city, with Chongqing, Chengdu, Zhengzhou, and Wuhan averaging around $7000 (USD)/FEU [19]. Table 7 shows information on the freight cost structure and subsidy of the five major CRE routes.

According to China Railway, some 40 cities and 90 routes have established CRE operations with CRE operating platform companies [40]. Many sections of these operating routes, however, overlap each other, making it difficult to achieve economies of scale from a big-picture perspective and promoting excessive competition between regions. Moreover, the subsidies that are used to attract cargoes put excessive fiscal pressure on local governments. Finally, the relatively high levels of subsidies to shippers undermines rational competition and has made sustainable growth difficult [1].

Within the major regions in China, CRE service is regarded as a means of achieving regional economic development and realizing the five connectivity strategies prioritized by the BRI. This motivates them to aggressively make subsidies available as a means of pursuing quantitative growth of the CRE [6]. For example, when cargo is transported by the CRE from Chongqing to Amsterdam, subsidies of about 62% or more of the total transport cost are provided. This route attracts cargo, as it allows for a two-week reduction in transit time and with almost no difference from maritime freight rates linking the inner and outer ports of the Yangtze River [40]. However, where, as here, competitiveness can only be achieved with subsidies, it is obviously unsustainable. However, as things currently stand, freight competitiveness would disappear immediately without them.

In the early stages of the CRE's development, financial aid by a local government may be justified to protect and encourage a nascent rail transport industry. However, the CRE subsidy system is heavily distorting the current market structure in transportation and logistics. The central government has accordingly placed limits on subsidies, capping them at 50% of the total cost of rail transportation in 2018, and requiring further 10% reductions every year. [46]. However, local governments have been offsetting these central-government-mandated reductions by providing other forms of indirect support to CRE operators, creating a situation in which it will be difficult for these cities to effectively exit their support for the block train system. In the meantime, the full extent of support will remain opaque and incomplete [16].

Accordingly, it is urgent that duplicated CRE routes be readjusted and CRE operators be compelled to develop genuinely subsidy-free operation structures that increase their competitiveness, as opposed to merely relying on quantitative growth as a success metric.

**Table 7.** Freight cost structure of the five major CRE routes.

| Division | Yuxinou | Rongou | Zhengou | Hanou | Sumanou |
|---|---|---|---|---|---|
| Destination | Duisburg | Lodz | Hamburg | Pardubice | Warsaw |
| Total distance (Km) | 11,179 | 9826 | 10,214 | 10,100 | 11,200 |
| Chinese broad-gauge distance (Km) | 4137 | 3511 | 3422 | 2918 | 3256 |
| European broad-gauge distance (Km) | 5692 | 5692 | 5692 | 5692 | 7739 |
| European standard-gauge distance (Km) | 1350 | 623 | 1100 | 1490 | 205 |
| Period (days) | 12–14 | 12–14 | 11–12 | 14 | 14 |
| Frequency (trains/week) | 2–3 | 1 | 2–3 | 1–2 | 1 |
| Chinese freight cost, USD/FEU·km (standard gauge) | 0.6 | 0.6 | 0.6 | 0.6 | 0.6 |
| Foreign freight cost, USD/FEU·km (broad-gauge) | 0.694 | 0.694 | 0.694 | 0.694 | 0.413 |
| Foreign freight cost, USD/FEU·km (standard gauge) | 0.704 | 2.73 | 0.864 | 1.713 | 3.415 |
| (un) Loading cost, (USD/FEU) | 1000 | 1000 | 1000 | 1000 | 1000 |
| Freight forwarding cost (USD/FEU) | 1800 | 1800 | 1500 | 1800 | 1500 |
| Subsidies [a] (USD/FEU) | 6400 | 7000 | 7400 | 5600 | 1000 |
| Freight costs (USD/FEU) | 10,182.85 | 10,557.64 | 9453.85 | 11,053.42 | 8349.88 |
| Freight costs (considering subsidies) (USD/FEU) | 3782.85 | 3557.64 | 2053.85 | 5453.42 | 7349.88 |
| Freight costs (balanced cargo flow), (USD/FEU) | 7721.9 | 7971.76 | 7135.9 | 8302.28 | 6399.92 |
| Freight costs (balanced cargo flow considering subsidies), (USD/FEU) | 1321.9 | 971.76 | 2135.9 | 2702.28 | 5399.92 |

[a] The value of subsidy to Zhengou equals to the gap between the CRE freight cost and the seaborne shipping cost. Source: Jiang et al. [19]. p. 195.

### 4.3. Expanding CRE Demand and Balancing Round-Trip CRE Cargoes

Maritime transport is means of international cargo transport through which it is most possible to achieve economies of scale through single shipments of large amounts of low-value-added cargo. Shipping also benefits from a relatively clear market position that is not sensitive to transit time. Air transport is targeted towards high-value-added cargoes that justify the expensive freight costs. In comparison, the CRE does not yet have a clear market niche [16]. The absence of clear target cargoes is a serious obstacle to the natural emergence of stable cargo volumes. In this context, the competition between regions to attract products that have very low added value, including daily necessities, is hindering the sound development of the CRE.

As a land bridge, the CRE provides a clear intermediate transport alternative that complements the advantages and disadvantages of sea and air transportation between China and Europe. Effective CRE operation requires clarity as to the profitability of possible cargoes, considering cost, price, cargo type, and other special circumstances. As CRE operators cannot indefinitely rely on government subsidies, it is imperative that they establish their own revenue operating model and develop marketing strategies to improve the efficiency, economics, and safety of the CRE transportation.

To increase demand for the CRE services, governments will also need to attract business interested in multimodal transport, specifically carto from Japan and Korea. Recently, shipping and logistics companies have expanded their use of the Eurasian railroad network as a multimodal transportation route. Moreover, as the COVID-19 pandemic has made it difficult to secure shipping, interest in the block train services offered by Russia and China are been increasing. CRE operators and the Chinese government should secure new CRE cargo for transport on the TCR by actively developing multimodal transport routes with shipping companies. In addition, new strategies to expand CRE cargo by combining train ferry and the CRE with South Korea, China's major trading partner.

The imbalance between export cargo volume from China and import cargo volume from Europe remains a challenge [25]. While the number of CREs departing from Europe have steadily increased since 2013, with the round-trip ratio improving to around 55:45 in 2019, the return rate of empty containers from Europe to China remains too high [47]. The major items imported into China are metals and appliances, transportation equipment, and chemical products [21], and it is essential that optimal transportation capabilities that match the unique characteristics of each product be provided.

For CRE to grow at a stable rate in the future, it must be made competitive in terms of safety, efficiency, and economic feasibility. This means increasing demand for CRE services and reducing risk factors associated with CRE-linked hardware and software. Chinese

central and local governments should first focus on hardware, including the continued building and improvement of railroad networks. China–Europe transportation routes have been hampered by inefficiencies and risks to safety in transportation due to differing levels of infrastructure at each pass-through point. Second, policy support should continue, rather than solely relying in direct financial support and customs clearance cooperation with BRI countries. Of course, the central and local Chinese governments should continue to promote cooperation for CRE operation with the BRI countries and regions.

The government should also induce and help CRE operators enhance their competitiveness. Moreover, CRE operators should work independently to identify opportunities to improve competitiveness and efficiency of transportation in Eurasia, and actively market themselves to attract sea–land multimodal transport cargoes.

## 5. Conclusions

The CRE is China's block train transportation service, and it operates regularly according to fixed trains, routes and schedules to transport containers between China, Central Asia, and Europe. CRE transport services have grown rapidly, and are regarded as critical logistics infrastructure to the Silk Road economic belt envisioned by the BRI. In March 2015, "Vision and Actions on Jointly Building Silk Road Economic Belt and 21st-Century Maritime Silk Road"document, which was jointly announced by the NDRC, the MOFA and the MOFCOM, presented and emphasized the importance of the CRE. Since then, the CRE has continued to grow, and it now serves as a barometer of the BRI's performance [23]. In quantitative terms, the operation performance of the CRE has grown 800x since 2011, as of 2019. Between China and Europe, the CRE connects 92 cities through 21 countries through an Eastern, Central, and Western corridor. Among these, the Rongou, as one of the CRE routes departing from Chengdu, and the Yuxinou, which starts in Chongqing, have seen the most active growth [3,5,48].

Recently, advanced shipping and logistics companies have begun expanding the use of the Eurasian railroad network as a multimodal transportation route that connects sea and block train services. Moreover, as the COVID-19 pandemic has made it difficult to secure shipping, the use of block train services by Russia and China has been increasing. Despite the expanding role and potential importance of the block train system in Eurasia, few academic studies have been undertaken on the relevant CRE policies and status.

This study comprehensively reviewed relevant domestic and foreign academic research, reports and policies, press news articles, and policy proposals related to China's railroad and CRE-related policies pursuant to the BRI, as well as the operation status of the CRE. Moreover, this study identified some problems associated with the CRE's rapid growth and conducted a rudimentary study for the design and use of a multimodal transport network incorporating the block train service.

This study presented several known challenges to the sustainable operation of the CRE: (1) resolving cargo concentration and bottlenecks at the CRE's borders, (2) alleviating competition and subsidies between local governments, and (3) expanding the CRE demand and balancing round-trip CRE cargoes. Each of these challenges may be considered by future researchers interested in railway border-efficiency analysis; economic evaluation in the context of subsidies; the CRE's logistics hubs; route and network optimization; customs clearance platform with block chain design, realizing the balance of trade cargo between China and Europe; and CRE market creation.

This study may have policy implications for the Chinese government and overseas logistics companies and shippers. First, it may help logistics companies design optimal transport routes and logistics networks, and help shippers understand CRE services, ultimately leading to greater use of the CRE. Second, by showcasing a niche logistics market to overseas logistics companies, the study should help those firms develop entry strategies into the Eurasian railroad logistics business. From the perspective of Korea, if train ferries and TKR (Trans–Korean railways) are connected in the future, this study can serve as reference material for the preparation of policies that will embody Korea's role as

an origin and destination station. As a first attempt at academic English-language research on the CRE's status and challenge, the author hopes that the study will prompt increased interest in this emerging topic.

**Funding:** This research received no external funding.

**Institutional Review Board Statement:** Not applicable.

**Informed Consent Statement:** Not applicable.

**Data Availability Statement:** Not applicable.

**Conflicts of Interest:** The authors declare no conflict of interest.

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
