# Peer review of "The Current Status and Challenges of China Railway Express (CRE) as a Key Sustainability Policy Component of the Belt and Road Initiative"

_sustainability, doi:10.3390/su13095017_

Round 1

Reviewer 1 Report

The paper needs minor editing in terms of English and scientific writing style and some parts of the paper are misleading or have no logical order or flow. The paper must go through minor editing and proofreading process before resubmission.

The 2nd main limitation of the paper is that it lacks critical related work. The historical perspective of the BRI should be discussed as well as the relationship of BRI and China Railway are not discussed. The proposed study is not critically evaluated and compared to the related work/state-of-the-art and is not identified and discussed its drawbacks and limitations. Thus, it is not easy to assess the real contribution of the paper in the field and how much is efficient in the proposed study compared to related works. A clear assessment of the contribution of the authors when compared to existing approaches should be given.

A fundamental item that was not clear in the study is: how and what should be the parameters to measure the sustainability of BRI with relationship to China Railway. It becomes necessary to propose future actions for improvement and sustainability by considering the previous studies carried out.

Author Response

Comment : The paper needs minor editing in terms of English and scientific writing style and some parts of the paper are misleading or have no logical order or flow. The paper must go through minor editing and proofreading process before resubmission.

  • Respond  : Thank you for your suggestion, I tried to reorganized and rewrite overall contents for more logical and consistent flow of this paper, and also corrected some non-grammatical parts and typos.

Comment : The 2nd main limitation of the paper is that it lacks critical related work. The historical perspective of the BRI should be discussed as well as the relationship of BRI and China Railway are not discussed. The proposed study is not critically evaluated and compared to the related work/state-of-the-art and is not identified and discussed its drawbacks and limitations. Thus, it is not easy to assess the real contribution of the paper in the field and how much is efficient in the proposed study compared to related works. A clear assessment of the contribution of the authors when compared to existing approaches should be given.

  • Respond : Many thanks for your comments. Specifically, the studies related to BRI and CRE were additionally reviewed in the literature review section and overall the contents of the literature review part was revised. Moreover, the limitations of the existing literature, the purpose and the contribution of the study were clarified.

Comment : A fundamental item that was not clear in the study is: how and what should be the parameters to measure the sustainability of BRI with relationship to China Railway. It becomes necessary to propose future actions for improvement and sustainability by considering the previous studies carried out.

  • Respond : Thank you very much for your constructive comments. In order to find the importance of China railway, CRE, under the BRI, the relationship between BRI and China railway were additionally explained in chapter 3.1. In addition, several alternatives were presented to propose future actions for the sustainable operation the CRE and development of the BRI, in chapter 4 based on status analysis of the CRE.
  • In addition to this, To help the readers understand, some figures have been added as follows :
  • Fig. 1. The spatial range of the BRI
  • Fig. 2. The five strategies for linking with BRI countriesFig
  • Fig. 3. The CRE routes by Eastern, Central, and Western
  • Fig. 4. The CRE related cities 

--> Please refer to the revision of this paper. Thank you very much for all of your comments on improving the quality of this paper. I have made every effort to revise this paper to accommodate all of your comments and suggestions.

Reviewer 2 Report

The author has deeply investigated the situation and the challenges for CRE. It is a good start to improve the efficiency and effectiveness of CRE for academic research. There are some general remarks for this paper:

  1. The purpose of this paper is to provide comprehensive review in English. However, it is difficult for readers with little knowledge of Chinese cities to follow the description. Especially it is necessary to use a unified description to explain the routes.
  2. Chapter 4 "Challenges for continuous the CRE development" is the most important part in this paper. The structure is not organized very well. For example, the problem of subsidies has been mentioned twice in 4.4 and 4.5. The connection between the information presented in Chapter 3 and the challenges listed in Chapter 4 is missed.
  3. It is important to point out, how could academic researchers to contribute in this field. E.g. to build a market-oriented analysis model, to solve the technical problem of freight railway system. A review to related research (not for CRE) would be helpful.

Some small remarks:

page 2 line 59: Some abbreviations need to be further explained. e.g. Yuxin-Ou (Chongqing-Xinjiang-Europe). It is too late to explain the term "Ou" in page 11 line 304. Also, the term is not used consistently (in page 3 line 137, the term "Chongqing-Europe" is used). It may be difficult for non-Chinese readers for understanding.

page 2 line 61: 92cities, a blank is missing.

page 7 line 211: typo "This The CRE is an important ..."

page 7: table 2 is shown in this page, however, it was mentioned in page 13. It is suggested to move table 2 to part 4.4

page 7: The name of routes are confusing. In the beginning of section 3.2, three routes (east, middle, west) are defined. However, the term "easten, central, westen" are used to explain them. In section 3.3, other routes (e.g. Xiang-ou) are mentioned. It is hard to follow for readers who are not familiar with the cities.

page 8: Would it be possible to use a map instead of table 3 to describe the routes?

page 8: What's the reason that in Table 4 the number of trains (2388) at Alashankou in 2018(1-11) is smaller than the number in Table 5 (1387+910)?

page 11: table 8 typo "fright" should be "freight" 

page 12 line 323: What is the reason of "lack of trains in border ports"? Usually the train is not starting from the ports. Please explain it. 

page 12 line 328: What is the meaning of "Yuxin-Ou /Yixin-Ou of China and Madrid of Spain"?

Author Response

Comment: The author has deeply investigated the situation and the challenges for CRE. It is a good start to improve the efficiency and effectiveness of CRE

  • Respond : As pointed out, some figures were added as follows to help readers better understand the Chinese cities and CRE routes :
  • - Fig.1. The spatial range of the BRI
  • - Fig.2. The five strategies for linking with BRI countries
  • - Fig.3. The CRE routes based on OD (Origin and Destination)
  • - Fig.4. The CRE related cities

Comment: Chapter 4 "Challenges for continuous the CRE development" is the most important part in this paper. The structure is not organized very well. For example, the problem of subsidies has been mentioned twice in 4.4 and 4.5. The connection between the information presented in Chapter 3 and the challenges listed in Chapter 4 is missed.

  • Respond : The contents of Chapters 3 and 4 were revised more consistently. As suggested, contents of Chapters 3 and 4 were restructured for a consistent and logical flow. In particular, first, the title of chapter 4 was revised to “Challenges for sustainable CRE development.” Second, chapter 4 has been rewritten by revising the previous version to the following three challenges in order to resolve the duplicated problems in the contents based on the status analysis of the CRE in Chapter 3.

- 4.1. Resolving cargo concentration and bottlenecks at the CRE borders

- 4.2. Alleviating competition and subsidies between local governments

- 4.3. Expanding the CRE demand and Balancing round-trip CRE cargoes

Comment: It is important to point out, how could academic researchers to contribute in this field. E.g. to build a market-oriented analysis model, to solve the technical problem of freight railway system. A review to related research (not for CRE) would be helpful.

  • Respond : Chapter 4 included some alternatives for the sustainable development of the CRE operation under the BRI, and several agendas were proposed for future academic research in the conclusion.

○ Some small remarks:

Comment : page 2 line 59: Some abbreviations need to be further explained. e.g. Yuxin-Ou (Chongqing-Xinjiang-Europe). It is too late to explain the term "Ou" in page 11 line 304. Also, the term is not used consistently (in page 3 line 137, the term "Chongqing-Europe" is used). It may be difficult for non-Chinese readers for understanding. page 2 line 61: 92cities, a blank is missing.

  • Respond : I tried to unify the CRE names in the contents and table. And as mentioned above, some figures were added to help readers better understand the CRE routes and cities.

Comment : page 7 line 211: typo "This The CRE is an important ..."

  • Respond : I corrected this typo.

Comment : page 7: table 2 is shown in this page, however, it was mentioned in page 13. It is suggested to move table 2 to part 4.4

  • Respond : The order and position of the tables were readjusted according the contents of the revision.

Comment: page 7: The name of routes are confusing. In the beginning of section 3.2, three routes (east, middle, west) are defined. However, the term "easten, central, westen" are used to explain them. In section 3.3, other routes (e.g. Xiang-ou) are mentioned. It is hard to follow for readers who are not familiar with the cities.

  • Respond : As mentioned above, I tried to unify the terms related CRE routes to avoid confusion these terms, and also Figures 3 and 4 are presented to help readers better understand CRE routes and cities.

Comment : page 8: Would it be possible to use a map instead of table 3 to describe the routes?

  • Respond : As you proposed, I presented the Figure 3 describing the CRE routes.

Comment : page 8: What's the reason that in Table 4 the number of trains (2388) at Alashankou in 2018(1-11) is smaller than the number in Table 5 (1387+910)?

  • Respond : Unfortunately I wasn't aware of this issue you pointed out. I tried to find this reason, however, I failed to find out the reason and alternative data. Therefore, I deleted the table 4 to avoid confusion.

Comment : page 11: table 8 typo "fright" should be "freight" 

  • Respond : I corrected it.

Comment : page 12 line 323: What is the reason of "lack of trains in border ports"? Usually the train is not starting from the ports. Please explain it. 

  • Respond : In order to avoid confusion of terms, 'border port' was modified to 'railway border'.

Comment : page 12 line 328: What is the meaning of "Yuxin-Ou /Yixin-Ou of China and Madrid of Spain"?

  • Respond : I modified it to “Yiwu or Chongqing to Madrid in Spain” in the 4.1 section.

Respond : In addition to what you proposed and suggested, the studies related to BRI were additionally reviewed in the literature review section, as well as the limitations of the existing literature, the purpose and the contribution of the study more clarified by rewriting. After all, this study has been revised based on an overall review for a logical and consistent flow. I have made every effort to revise this paper to accommodate all of your comments and suggestions.

Please refer to the revision of the paper. Thanks again for your constructive and valuable comments.

Round 2

Reviewer 1 Report

The manuscript is now improved a lot as per the suggestions provided. However, the manuscript still lacks the quality of the English Language as per the standard of the Journal. Therefore, it is advised to use the Third-Party Language editing service and provide the certificate of editing. 

Author Response

As advised, this manuscript received editing services to improve the quality of English at the Jeonbuk National University Writing Center.

I deeply appreciate all of your helpful comments and suggestions, I tried to apply and incorporate all into the paper. Thank you very much.

Reviewer 2 Report

The revised version is now improved significantly. The comments and the issues from the last review has been addressed. It is a good review on CRE for sustainable development. I look forward to see further contributions from the authors in this direction.

Author Response

Finally, this manuscript received editing services to improve the quality of English at the Chonbuk University Writing Center.

I deeply appreciate all of your comments and acceptance to publish this paper. Thank you very much.